# An objective, automated and robust scoring using fluorescence optical imaging to evaluate changes in micro-vascularisation indicating early arthritis

Lukas Zerweck[1,3]*, Michaela Köhm[2,3]*, Phuong-Ha Nguyen[1,3], Gerd Geißlinger[2,3,4], Frank Behrens[2,3,5], Andreas Pippow[1,3]

1 Fraunhofer Institute for Applied Information Technology FIT, Sankt Augustin, Germany, 2 Fraunhofer Institute for Translational Medicine and Pharmacology ITMP, Frankfurt am Main, Germany, 3 Fraunhofer Cluster of Excellence Immune-Mediated Diseases CIMD, Frankfurt am Main, Germany, 4 Clinical Pharmacology, University Hospital Goethe-University, Frankfurt am Main, Germany, 5 Rheumatology, University Hospital Goethe-University, Frankfurt am Main, Germany

* lukas.zerweck@fit.fraunhofer.de (LZ); michaela.koehm@itmp.fraunhofer.de (MK)

**Data Availability Statement:** 688 files are available from the zenodo.org database. The link to the data

## Abstract

Fluorescence optical imaging technique (FOI) is a well-established and valid method for visualization of changes in micro vascularization at different organ systems. As increased vascularization is an early feature of joint inflammation, FOI is a promising method to assess arthritis of the hands. But usability of the method is limited to the assessors experience as the measurement of FOI is semi-quantitative using an individual grading system such as the fluorescence optical imaging activity score (FOIAS). The goal of the study was to automatically and thus, objectively analyze the measured fluorescence intensity generated by FOI to evaluate the amount of inflammation of each of the subject's joints focusing on the distinction between normal joint status or arthritis in psoriatic arthritis patients compared to healthy volunteers. Due to the heterogeneity of the pathophysiological perfusion of the hands, a method to overcome the absoluteness of the data by extracting heatmaps out of the image stacks is developed. To calculate a heatmap for one patient, firstly the time series for each pixel is extracted, which is then represented by a feature value. Secondly, all feature values are clustered. The calculated cluster values represent the relativity between the different pixels and enable a comparison of multiple patients. As a metric to quantify the conspicuousness of a joint a score is calculated based on the extracted cluster values. These steps are repeated for a total number of three features. With this method a tendency towards a classification into unaffected and inflamed joints can be achieved. However, further research is necessary to transform the tendency into a robust classification model.

repository is: https://doi.org/10.5281/zenodo.5705208.

**Funding:** The authors declare that there was funding from the Fraunhofer Excellence Cluster for Immune mediated diseases (https://www.cimd.fraunhofer.de/en.html). The funders had no role in study design, data collection and analysis, decision to publish, or preparation of the manuscript.

**Competing interests:** The authors have declared that no competing interests exist.

## Introduction

Musculoskeletal (msk) pain, especially joint pain, is a very common symptom potentially leading to chronic pain syndrome and impact of function if a proper treatment is not initiated. Mechanical pain is the most common reason for joint complains, but pain can also be caused by inflammation associated to immune mediated diseases. Different types of arthritis can be identified, from which rheumatoid arthritis is the most frequent one, affecting approximately 1% of the adult European population [1]. The differentiation between the mechanical and inflammatory cause for joint pain is of high importance to avoid structural damage of the joints by choosing a specific treatment.

In clinical practice, arthritis is diagnosed by clinical examination of swollen and tender joints by rheumatologists. Additionally, imaging techniques can be used to quantify the amount of inflammation. Imaging methods such as ultrasound with power-doppler technique are widely used in Europe but limited to the assessor's experience, high inter-reader variability and to its time consuming procedure when used for assessment of all joints of the msk-system. MRI can be used as an alternative to ultrasound with a higher sensitivity and specificity, but has limitations due to its accessibility and expensiveness. In early stages of arthritis, swelling may not be evident whereas inflammation can be detected sensitively using imaging methods of the joints (e.g. ultrasound or MRI). Fluorescence-optical imaging is an indocyanine green (ICG) tailored method to visualize micro vascularization of the hands [2–4]. It might be of high value for detection of early arthritis when clinical examination may not lead to a clear diagnosis as it is well-tolerated by the patients and easily accessible for the physicians. By now, FOIAS is assessed semi-quantitatively and depends on the reader's experience in evaluation of the film [2–4] although fluorescence intensities are measured quantitatively.

The aim of this work is to overcome the semi-quantitative assessment method and replace it with an objective, reproduceable and quantitative assessment system for the FOI images. Comparing the per se signal between subjects is not trivial, since patients have very heterogeneous perfusion of the hands. Therefore, defining threshold values describing the health status of each joint, using the raw data, is impossible. The suggested method overcomes the absoluteness of the measured data by calculating heatmaps visualizing conspicuous pixels regarding different feature values applied in a cohort of patients with psoriatic arthritis compared to healthy volunteers.

## Related work and comparison to the proposed method

Previously the value of fluorescence optical imaging (FOI) as a diagnostic tool to detect any kind of joint inflammation e.g. synovitis was shown [5, 6]. Furthermore its specificity and sensitivity in comparison to 1.5 T MRI [7], 3 T MRI [8], ultrasonography in grey scale and power Doppler mode [6, 7] and clinical examination [6–8] were analysed and described in detail. Additionally, there were attempts to automatically classify patients as well as single joints into healthy and inflamed [9, 10]. In [9] the time series signal is extracted from the joint areas and the health status distinguished by the time series. Even though Dziekan *et al.* achieved a distinction between affected and unaffected joints within one patient diagnosed with rheumatoid arthritis, this approach shows a lack of comparability between patients. This incomparability is caused by physiological variability and inter-individual differences of the characteristics of the microvascularisation. In this work heatmaps based on three features extracted from the image stack are generated to overcome this challenge. In [10] a principal component analyses (PCA) over temporal subsequences of the image stack are performed. With this approach a promising result was achieved to distinguish between "healthy or mild synovitis and moderate or severe synovitis" [10, p.14]. However, a lack of distinction between unaffected and mild synovitis was

observed. Nevertheless, there is a high medical need for distinguishing between not and mild inflamed joins.

## Patient group

All included patients were diagnosed with active psoriatic arthritis (PsA). Additionally, 12 healthy volunteers without any joint complains were recruited and are used as a control.

For 163 patients an assessment of clinical signs of PsA (swollen and tender joint count and its evaluation by a rheumatologist) is available.

FOIAS was measured in 91 patients using grading according to ICG-distribution [2–4]. Increase of vascularization in the different ICG-distribution phases was rated by an experienced central reader on a 0–3 scale. Here, a score of 0 represents no visible conspicuousness, while score 3 indicates high visibility.

The 12 healthy volunteers were selected with regards to the following in- and exclusion criteria: (a) missing musculoskeletal complaints on the hands, (b) missing diagnosis of joint diseases, (c) missing comorbidities with focus on diseases that go ahead with joint diseases, (d) missing contraindications to use indocyanine green as colour agent for FOI examinations. Due to these criteria, the 12 volunteers are not affected by PsA and can act as a control group.

For an unbiased investigation of the proposed method, no further subgroups are formed. Data were analyzed in a blinded manner. Therefore, all additional parameters, which can influence the perfusion of the hands e.g. gender, age, weight, smoking habit, surrounding temperature etc. are not taken into account.

For performance of the analysis of the pseudonymized clinical and imaging data, patients / volunteers were recruited from two prospective non-interventional studies (one with inclusion of patients diagnosed with psoriasis or psoriatic arthritis and one with inclusion of healthy volunteers without complaints in the joints of the hands). The study fulfilled Good Clinical Practice Guidelines and all patients / volunteers provided signed informed consent for inclusion in the studies and agreed to the usage of their data for research purposes. All clinical procedures were performed according to study protocol, which received ethical approval from the ethics committee of the University Hospital Frankfurt a. Main, Germany. All patients / volunteers were fully capable to give informed consent for participation in a study.

## Data acquisition

The proband places her/his hands into a preformed template beneath the camera and inside the Xiralite X4 machine. Then, a run to acquire the images is started. To reduce the influence of the environment on the data, the room's windows are covered and the lights are switched off. After 10 s the prepared colour agent indocyanine green (ICG) is injected into one of the proband's arms with a dose of 0.1 $^{mg}$ICG/kg$_{body\ weight}$. The measurement takes six minutes in which an image stack of 360 pictures is taken (one picture per second). The colour agent is the fluorescent substance and thus essential for each measurement. Fig 1 visualizes the measured data.

## Method

All described methods were implemented using Python.

To calculate the score for each joint in the hands, two separate main processes have to be carried out and combined to a final result. Firstly, heatmaps based on features extracted from the image stack are generated. Secondly, the joints' positions and sizes are determined. Finally, the heatmaps are investigated and evaluated in the defined joint areas. However, this work

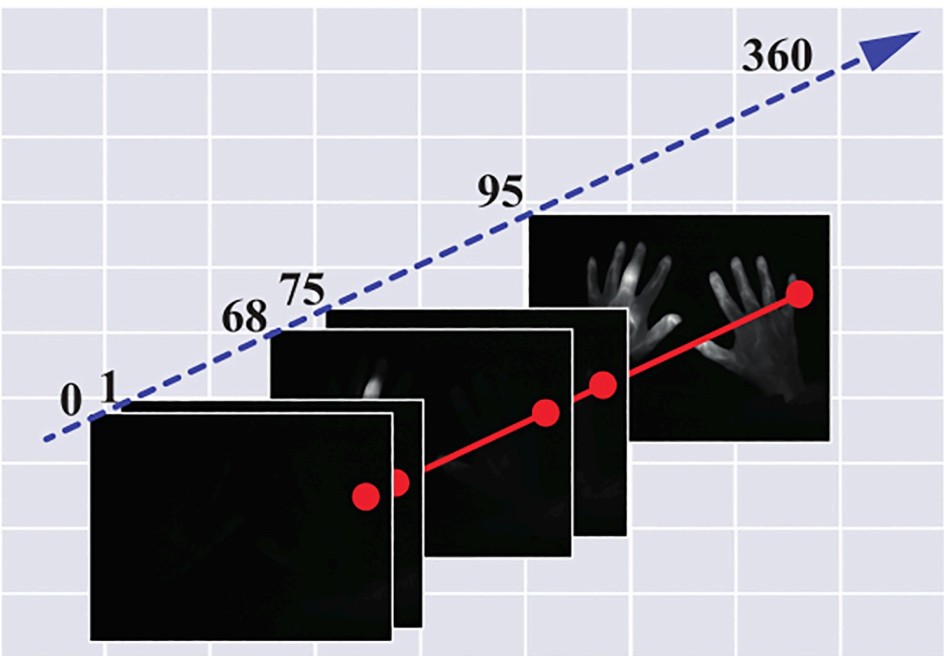

**Fig 1. Image stack (for visualization purposes).** The red line visualizes the extraction of one pixel time series.

mainly focuses on the heatmap calculation and final evaluation. Thus, the segmentation of the joint areas is only described briefly.

## Binary mask extraction

Since a binary mask is used in the segmentation as well as the heatmap calculation (section Joint Segmentation and Generating the heatmaps based on the image stack), the mask extraction from the image stack is crucial for the final result. To calculate this mask a maximum image is generated based on the 360 image time series. Each pixel of the maximum image corresponds to the highest value of the according pixel time series. After enhancing the contrast of the maximum image, a bounding box around the two hands is defined and the "GrabCut" algorithm [11] is used to get the binary mask. The bounding box is defined by the spiking pixel values within the hand region. The different steps are visualized in Fig 2A–2C.

Using the method with the maximum image guarantees that the final mask includes possible movement artefacts. The tracking of the exact movement is lost, however the proposed heatmap approach is robust against smaller movements.

## Joint segmentation

Two methods are combined to determine the joints' positions, since none of them performs well on all joints. One algorithm is based on classic segmentation techniques combined with anatomical hand proportions [12]. In this approach, based on the extracted binary mask (see section Binary mask extraction), the fingers are detected and their length calculated. Since previous work have set the joints' locations in proportion to their finger and the corresponding finger length, the position can be estimated. This method works best for the Proximal (PIP) and Distal (DIP) Interphalangeal joints.

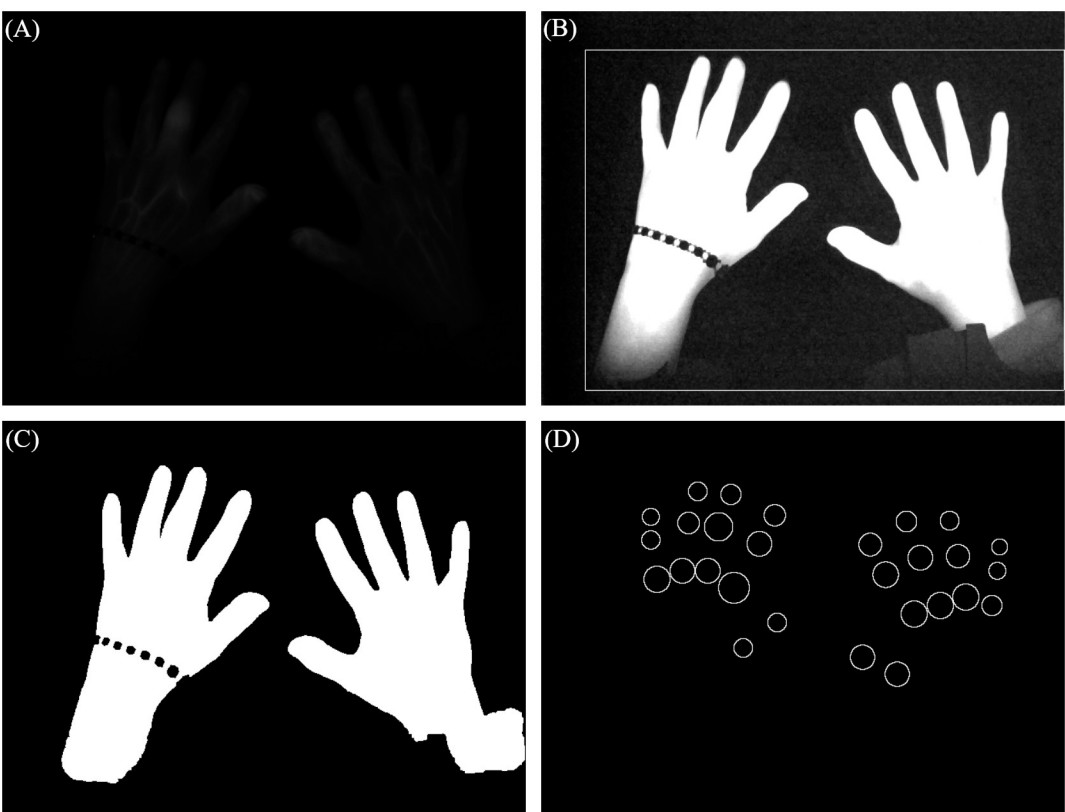

**Fig 2. Steps of the binary mask extraction and result of the joint region calculation.** (A) Extracted maximum image. (B) Enhanced image with bounding box. (C) Final binary mask. (D) Final joint areas.

In the second method the joints' locations are calculated with the pretrained neural network OpenPose [13]. With this approach all joint positions are calculated at once. However, only the precision for the Metacarpophalangeal (MCP) and Interphalangeal (IP) joints increases in comparison to the classical approach and therefore, are used for further analysis.

All joint regions are defined as circles. The radius is either defined over the smallest distance between the defined joint location and the finger edge (IP, DIP, PIP) or the smallest distance to the next joint location (MCP II-V). The radius of the joint MCP I is equal to the diameter of the thumb (the result is visualized in Fig 2D).

## Generating the heatmaps based on the image stack

The time series has a characteristic shape as described in [9]: a steep ascent, followed by a slow descent after reaching the peak (compare Fig 3A). Even though Fig 3A visualizes the average pixel value of one image at certain time points, each pixel within the signal region (hand region) follows a similar shape.

In the proposed approach, the time series is extracted at each pixel (an example is shown in Fig 3). Each time series is used to extract three features. Thus, each pixel is represented by three independent values. All values of one feature are collected in one set of data. Therefore, three independent sets of data are created, in which each value represents one pixel. The three features are the amplitude $\Delta I = I_{\max} - I_{\min}$, the mean pixel value during the increasing time $\bar{I} = \frac{\sum_{i=a}^{b} I_i}{b-a}$ ($a$ denoting the frame for $I_{min}$ and $b$ the frame for $I_{max}$) and the maximal gradient

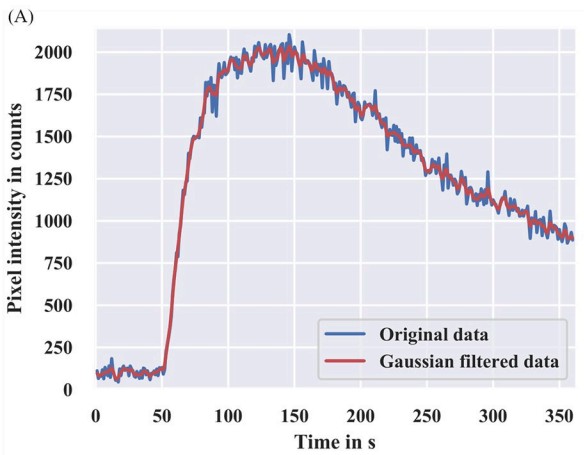 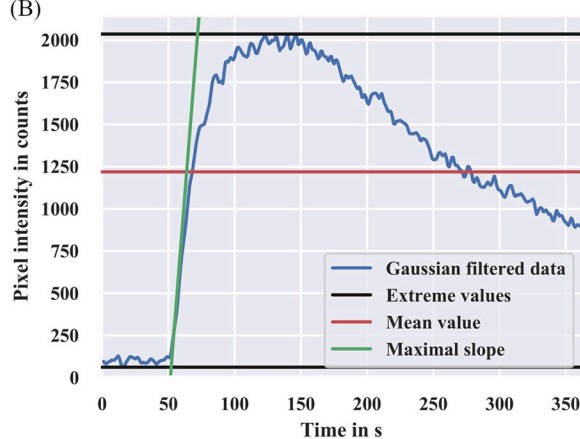

**Fig 3. Smoothing and feature extraction of one representative pixel time series.** (A) Gaussian filtering of the pixel time series. (B) Feature extraction of the pixel time series.

during the increasing time $\max\{I_{i+1} - I_i\}$ with $i \in \{\min \leq i \leq \max -1\}$ (compare Fig 3B). Since, the data is discrete and thus the difference between the current and the next point is calculated. The time point in which the calculated difference has the highest value is defined as maximal gradient.

To create the heatmaps from the three data sets, k-means clustering is performed on each of the data sets. Therefore, every pixel gets a cluster assigned, which is represented by a colour. The corresponding pixels in the (still empty) heatmap is set to the assigned pixel color. For k-means clustering the number of centroids has to be predefined. Since the best number of centroids is unknown, k-means clustering is performed four times choosing $k$ as 3, 5, 7 and 9. Thus, for each of the three features 4 heatmaps are created (an example is shown in Fig 4).

**Additional steps to lower the noise in the heatmaps.** In the process of generating the heatmaps two additional steps are performed to decrease the noise in the heatmaps. Firstly, to suppress any kind of interference with the background signal, the binary hand masks (see section Binary mask extraction) are applied and the background is completely set to 0. Secondly, after extracting the time series for one pixel, the data is smoothed by applying a Gaussian filter (standard deviation $\sigma = 1$). Therefore, outliers have less impact on the outcome (compare Fig 3A).

## Heatmap interpretation

Fluorescence optical imaging visualizes the current distribution of the colour agent ICG. Joints suffering from an inflammation show a higher perfusion [14] with new formation of vessels especially in PsA. The assumption is that inflamed joints have a higher signal in comparison to non-inflamed hand regions. Due to the higher perfusion into the joints the signal increases faster than in non-inflamed hand regions.

The three features are chosen to represent these properties of the perfusion. The amplitude and mean feature correspond to the amount of blood and the slope to the streaming speed. Even though the amplitude and mean represent the same physical property, the mean feature includes more of the time dependency of the data. Especially, fluctuations or a decline of the slope is not affecting the amplitude value.

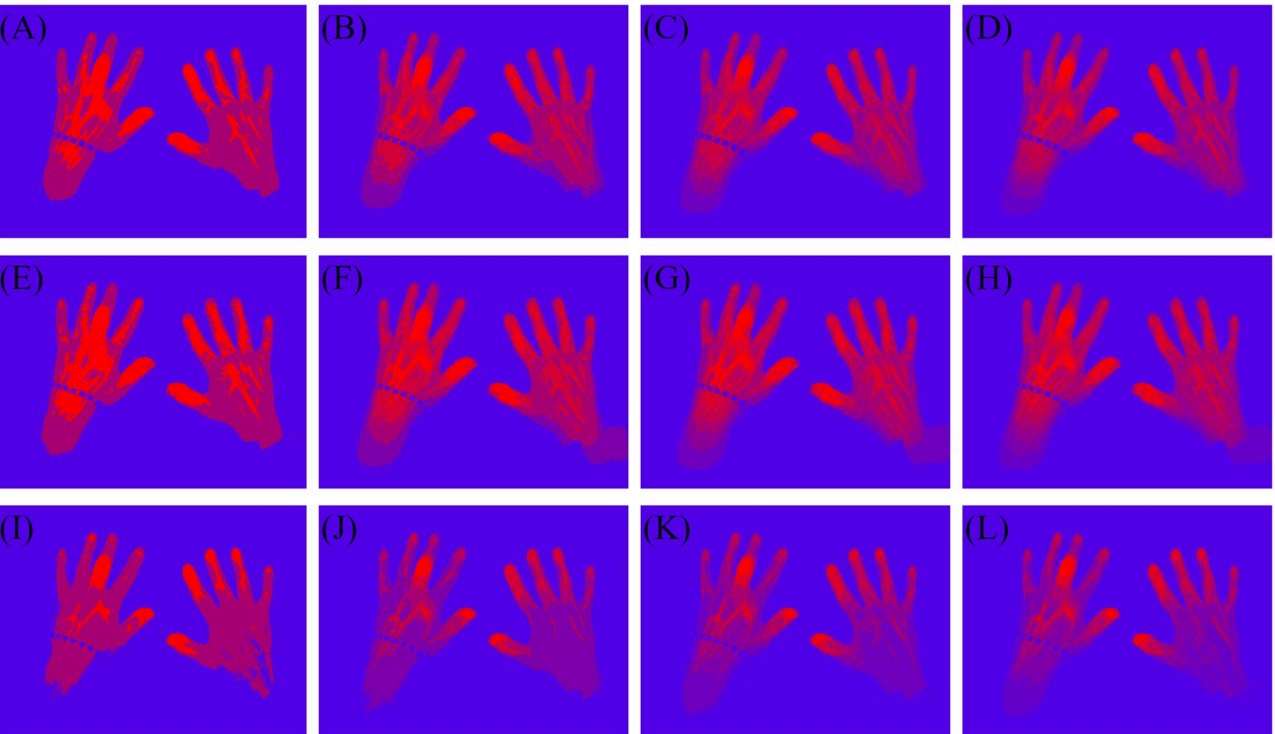

**Fig 4. Calculated heatmaps for the three features.** The row specifies the feature and the column the amount of cluster used creating the heatmap. For this example the slope feature (third row) emphasise the different health conditions of the joints. In the slope three cluster heatmap the right DIP II joint is assigned to the same cluster as the left PIP III joint. The slope seven and slope nine cluster heatmap represent the actual inflammatory condition better and show a clear distinction between right DIP II and left PIP III joints. For different examples each feature shows a different classification power. (A) 3 cluster amplitude. (B) 5 cluster amplitude. (C) 7 cluster amplitude. (D) 9 cluster amplitude. (E) 3 cluster mean. (F) 5 cluster mean. (G) 7 cluster mean. (H) 9 cluster mean. (I) 3 cluster max slope. (J) 5 cluster max slope. (K) 7 cluster max slope. (L) 9 cluster max slope.

## Scoring the joints based on the heatmaps

In the final step the heatmaps are investigated and evaluated within the defined joint areas (compare Fig 5). At first a value is assigned to each pixel within the joint area depending on its color. The lowest cluster (blue) (compare any picture in Fig 4) is represented by 0, while the highest cluster (red) is represented by the number of clusters minus one. All clusters in between get the corresponding number assigned. For example in a seven cluster heatmap blue pixels add 0 to the joint score, while red pixels add 6 to the joint score. To calculate the final score $S$ for one joint $j$, all assigned pixel values $p$ are added and divided by the number of pixels $n$ and the number of clusters $m$.

$$S_j = \frac{\sum_{i=1}^{n_j} p_{j,i}}{n_j \cdot m} \tag{1}$$

Due to the new idea of evaluating the data, the calculated scores do not have an explanatory power without setting them into context. However, due to the explanations in section Heatmap interpretation the assumption is made that for all features the scores for unaffected labelled joints are in general lower than for affected labelled joints.

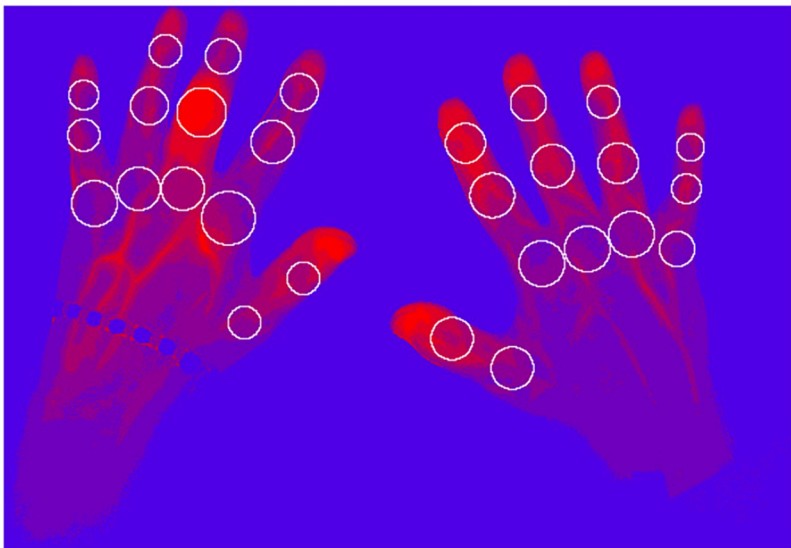

**Fig 5. Example for combining the extracted joint (compare Fig 2D) areas with a heatmap (compare Fig 4K).**

## Result

To evaluate the proposed method's eligibility to detect affected joints the calculated scores are connected to two different labels: the clinical label (swollen and tender joint assessed by clinical examination) and the FOIAS (described in section Patient group). The two different labels include three and four sub-categories.

- clinical label with the sub-categories: affected (swollen or tender), swollen or tender

- FOIAS with the sub-categories: affected (score higher 0), score = 1, score = 2 or score = 3

To visualize the statistical outcome of all sub-categories notched box plots are used. Each sub-category contains 9 box plots, which form 3 groups of 3. Each group visualizes the calculated score distribution for one feature (amplitude, mean and slope). In each group the left (red) box plot represents the scores for affected (PsA) and the box plot located in the middle (blue) the unaffected labelled joints for patients with a confirmed PsA. The third box plot on the right (purple) describes the distribution for healthy volunteers. Additionally, the mean value for each box plot is added as a black dot.

However, the results for the different amounts of cluster are not integrated into this visualization. Therefore, each figure not only represents one sub-category but also the amount of cluster (e.g. sub-category: swollen, cluster: 7). For clarity only the figures visualizing the results for $k = 7$ are embbeded into the manuscript. The remaining graphs for all other clusters as well as the mean and median values are visualized and summarized in S1–S6 Figs.

Finally, the predictive power of trained machine learning systems based on the three features and labelled by the clinical label as well as the FOIAS are investigated.

### Scoring results in comparison to the clinical labels

In section Patient group it is mentioned that for 163 patients with a confirmed PsA a clinical assessment (swollen or tender joints) is available, which correspond to $163 \cdot 28 = 4564$ assessed joints. 3824 joints were labelled as unaffected, 455 as tender and 285 as swollen. Furthermore, the 12 healthy volunteers result in 336 healthy labelled joints.

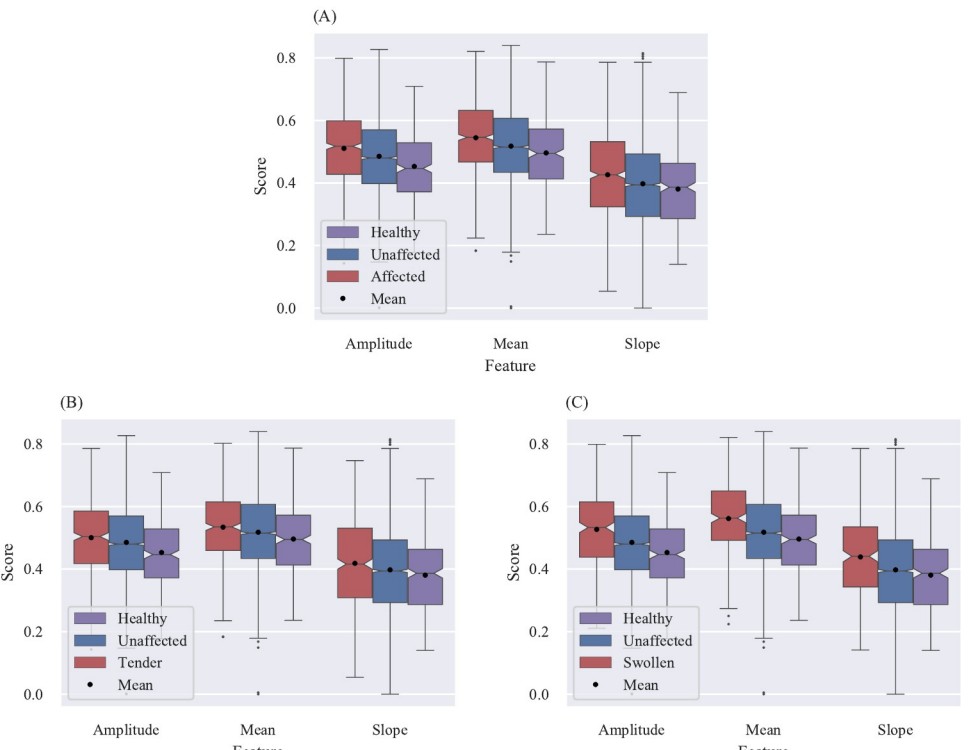

**Fig 6. Comparison between the calculated scores for unaffected and affected labelled joints based on clinical labelling (k = 7).** (A) Comparison between affected and unaffected labelled joints (sub-category: affected, k = 7). (B) Comparison between tender and unaffected labelled joints (sub-category: tender, k = 7). (C) Comparison between swollen and unaffected labelled joints (sub-category: swollen, k = 7).

In Fig 6 the results for the different sub-categories of the clinical label are summarized. The three graphs including the 9 box plots show all similar distributions of the calculated joint scores. Even though there is a big overlap between the unaffected and affected (tender, swollen, tender and swollen) distributions, a clear tendency towards a lower score in unaffected labelled joints can be observed. This observation is supported by the median and average scores. For 33 out of 36 groups (four cluster, three feature, three label) the notches comparing the distributions of unaffected and affected labelled joints do not overlap (Fig 6 and section S2 Fig). This suggests that the true medians of these distributions differ with a confidence of 95% [15]. The scores for the healthy volunteers are in general lower than the affected and unaffected score distributions.

However, for the tender labelled joints the notches of the healthy distribution overlaps with the notches of the unaffected and affected distributions for the slope feature choosing $k$ as 3, 7 or 9. Additionally, the healthy probands show an overall smaller scattering than the other two distributions.

For the swollen labelled joints no overlap between the healyth distribution and the swollen distribution among all clusters and features can be observed.

The three different sub-categories show different abilities to classify the joints. The distribution for tender joints (Fig 6B) shows the smallest and the swollen distribution (Fig 6C) the biggest difference between unaffected and affected joints. Since Fig 6A includes both, tender and swollen joints, it represents the average distribution.

## Scoring results in comparison to the experts evaluation of the FOI images performed by an experienced central reader using FOIAS [2–4]

For all calculations the maximal joint FOIAS score among all three phases is considered as the label value.

In section Patient group it is mentioned that for 91 patients a FOIAS assessment is available, which correspond to 91 · 28 = 2548 assessed joints. 2278 joints are labelled as unaffected, 102 as score = 1, 150 as score = 2 and 18 as score = 3. The healthy distribution is again formed from the 12 healthy volunteers.

In Fig 7 the results for the different sub-categories of the FOIAS are summarized. The graphs show, that with increasing FOIAS score the overlap between unaffected and affected labelled joints decreases and thus, the distinction between unaffected and affected increases (compare Fig 7–7D). The same tendency can be observed by comparing the medians and means of the distributions. Furthermore, besides a slight overlap for the notches of the slope feature in the score = 1 graph, no overlap between the affected and unaffected distribution can be observed. This suggests that the true medians differ with a confidence of 95%. The score distributions for the healthy volunteers are in general lower than the other two distributions. However, similar to the clinical assessment the notches of the unaffected and healthy distributions overlap for the slope feature. For some of the sub-category score = 1 the notches of the

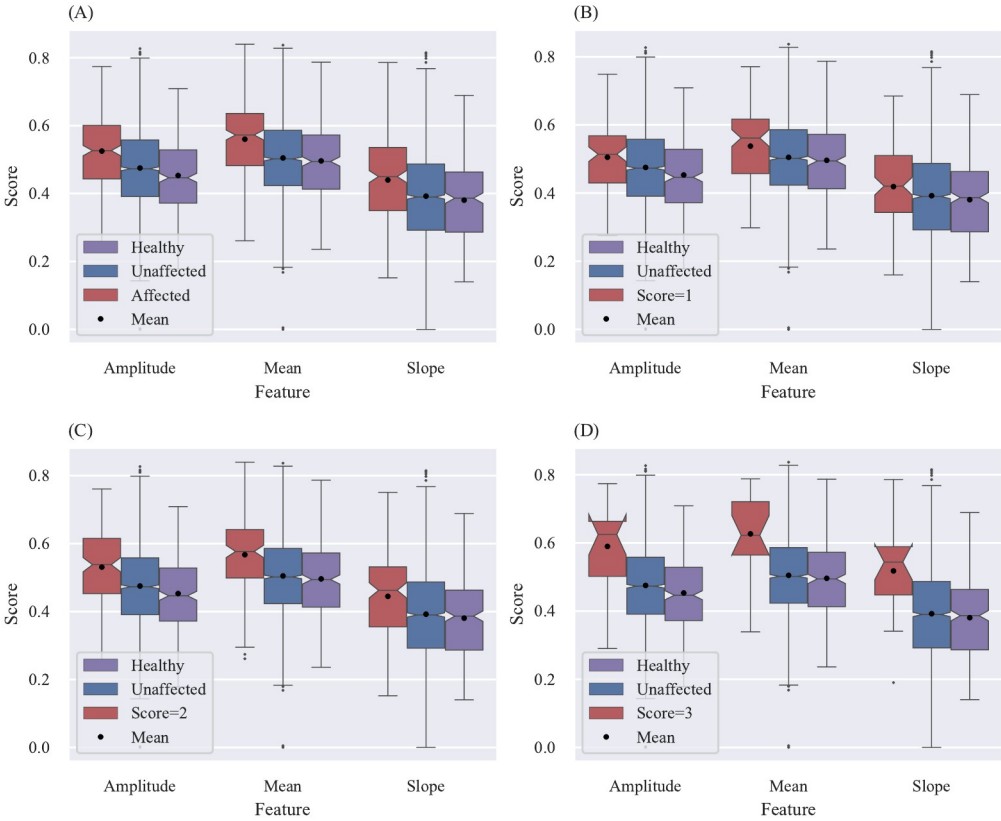

**Fig 7. Comparison between the calculated scores for unaffected and affected labelled joints using FOIAS (k = 7).**
(A) Comparison between affected and unaffected labelled joints (sub-category: affected, k = 7). (B) Comparison between score = 1 and unaffected labelled joints (sub-category: score = 1, k = 7). (C) Comparison between score = 2 and unaffected labelled joints (sub-category: score = 2, k = 7). (D) Comparison between score = 3 and unaffected labelled joints (sub-category: score = 3, k = 7).

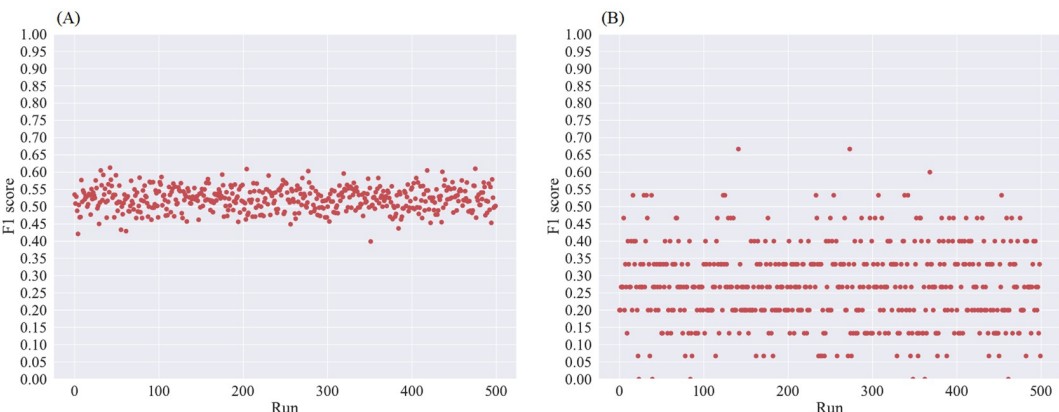

**Fig 8. F1 score results of the trained and tested random forest classifier using scores calculated with $k = 7$ clusters.** (A) Results for the 500 runs using the clinical label as ground truth. (B) Results for the 500 runs using the FIOAS label as ground truth.

affected and healthy distribution overlap as well. Due to the small sample size of score = 3 labelled patients, in same cases the confidence interval of the median extends the third quartile, which results in an unusual boxplot shape.

Fig 7A includes all joint scores with a FOIAS score higher than 0 and thus, represent the overall possibility of the suggested method to distinguish between unaffected and affected.

## Investigation of a trained machine learning system to predict the health status of a joint based on the calculated features

In order to investigate the predictive power of the three scores calculated for each joint to predict the heath status a random forest [16] classifier for the $k = 7$ scores is trained. The performance is independently investigated for the two different labelling methods (clinical labelling and FOIAS labelling). Since, both cases suffer from class imbalance the sizes of the overrepresented classes are cropped to the smallest class size. For the clinical label the class with the least amount of examples contains 740 data points and for the FOIAS label 18 data points. For both cases the data points of the overrepresented classes are picked randomly. Due to the cropping many data points remain unused. Therefore, the classifier has been trained 500 times for each labelling method.

Since for the clinical labelling a joint can be labelled as unaffected, swollen, tender or swollen and tender the classifier has been trained to distinguish only between unaffected or affected scores. Thus, a binary classification problem has to be solved. The calculated F1 scores range from around 40% to around 62% detection rate. The results are visualized in Fig 8A.

For the FOIAS labelling a 4 class (Score 0, 1, 2 or 3) classification problem has to be solved. The calculated F1 scores range from 0% to 66% detection rate. The results are visualized in Fig 8B.

## Discussion and conclusion

The calculated scores show the expected outcome. The scores for unaffected labelled joints are in general lower than the scores for joints labelled as affected. Furthermore, the distributions for tender joints show the smallest difference between unaffected and affected joints. This can be explained with the lowest comparability between patients, since the sense of pain is highly subjective. Additionally, the assumption is made that a swelling goes along with an

inflammation in the affected joint, which would lead due to the increased perfusion to a higher signal (compare section Heatmap interpretation). Since more joints are labelled as tender than labelled as swollen the average distribution (compare Fig 6A) is closer to the tender distribution (compare Fig 6B) than the swollen distribution (compare Fig 6C). Furthermore, the distributions for the FOIAS label shown in Fig 7 correspond to the expected outcome, since a higher FOIAS score is equal to a higher visibility in the images for the investigated joint.

Across all labelling methods, the score calculations based on the slope feature seem to lead to a higher overlap between unaffected and affected labelled joints. One explanation for the overlap is that in theory the finger tips and fingers are firstly visible in the image stack [17]. Therefore, k-means clustering will classify these regions as conspicuous, regardless of the perfusion of any other patient. This affect of the relative data is most dominant in the slope feature, since the blood streaming is not homogeneously distributed over the hand region. Furthermore, 64% of the joints are located within fingers, which emphasises the anatomical bias.

Wider notches of some distributions in comparison to other distributions, which in a few cases leads to an overlap of the notches, are caused by the smaller sample set.

Comparing the different numbers of clusters leads to the conclusion, that a higher number does not necessarily lead to a clearer distinction between the unaffected and affected score distribution. However, comparing the box sizes of the 3 cluster and 9 cluster box plots suggest that the variety within the distribution based on the 3 cluster heatmap is higher than the distribution based on the 9 cluster heatmap. Thus, choosing $k$ as 7 or 9 seems sufficient. The observed results indicate, that the presented idea can overcome the high inter-patient variability in the data, hence the majority follows the same tendency (compare Figs 6 and 7, S1–S6 Figs).

The approach to train a machine learning system based on the calculated feature scores does not lead to a robust classifier. The fluctuating results can be explained by the combination of overlapping scores and class imbalances as well as the features' properties. Due to the cropping and randomly picking of data points, the overlap of scores differs for each run, which leads to fluctuating results. Additionally, the data shows that an affected joint can have a high variety of score combinations along the different features. Therefore, an understanding of the causes for the heterogeneity within the data, the impact of non-disease related factors on the data and developing prepossessing steps to homogenise the data is crucial.

The comparison between unaffected and FOIAS score = 1 labelled joints is the most interesting, since this work is embedded into a project with the goal of the early detection of arthritis. Even though a high overlap between unaffected and affected labelled joints can be observed in Fig 7B, it also shows a clear tendency towards a classification into missing or mild arthritis. Therefore, the suggested objective, reproduceable and quantitative assessment system shows a promising first result and motivates further analysis.

## Future work

The presented method could lead to a sufficient diagnosis of arthritides such as PsA. However, there are limitations planned to be addressed in a following study.

### Non-disease related impact factors

To investigate the non-disease related impact factors and therefore, understanding the heterogeneity in more detail, a study among heathy volunteers is planned. The impact of factors like hand temperature, BMI, alcohol consumption and many more are planned to be addressed.

### Normalization of the images on the used machine

In the proposed approach the features values of all pixels in the image are compared with one another, regardless of their location within the image. However, the illumination of the Xiralite device is not homogeneous and thus, the pixels are technically incommensurable. The centre part of the image is in average brighter than the outer part. Therefore, taking a reference background image, capturing the light gradient in the device, and applying it to the raw images leads to a commensurable data set.

### Automated feature extraction

The heatmaps are calculated based on the extracted three time series features amplitude, mean during the increasing time and maximal slope. Even though, these three features already enabled to achieve a tendency for unaffected and affected scores, there could be features with a stronger classification power. With enough data a machine learning approach could choose better features and lead to better results.

### Normalization of the images on the proband

With the heatmap approach a method to overcome the heterogeneity of the data and to achieve an inter-proband comparability is suggested. However, only focussing on relative data can lead to unexpected outcomes. For example a completely healthy person could get in average a medium high score due to fluctuation within the data, even though all joints should lead to low scores. This phenomenon is already observable for the healthy probands. A process combining the absolute data with the relative data could lead to much clearer results.

### Definition of analysed joint areas

The conspicuous areas and the joint areas do not always match. Therefore, a method is needed to evaluate the heatmaps at the correct areas.

## Supporting information

**S1 Fig. Comparison between the calculated scores for unaffected and affected labelled joints based on clinical labelling (k = 3).** (A) Comparison between affected and unaffected labelled joints (sub-category: affected, k = 3). (B) Comparison between tender and unaffected labelled joints (sub-category: tender, k = 3). (C) Comparison between swollen and unaffected labelled joints (sub-category: swollen, k = 3).
(TIF)

**S2 Fig. Comparison between the calculated scores for unaffected and affected labelled joints based on clinical labelling (k = 5).** (A) Comparison between affected and unaffected labelled joints (sub-category: affected, k = 5). (B) Comparison between tender and unaffected labelled joints (sub-category: tender, k = 5). (C) Comparison between swollen and unaffected labelled joints (sub-category: swollen, k = 5).
(TIF)

**S3 Fig. Comparison between the calculated scores for unaffected and affected labelled joints based on clinical labelling (k = 9).** (A) Comparison between affected and unaffected labelled joints (sub-category: affected, k = 9). (B) Comparison between tender and unaffected labelled joints (sub-category: tender, k = 9). (C) Comparison between swollen and unaffected labelled joints (sub-category: swollen, k = 9).
(TIF)

**S4 Fig. Comparison between the calculated scores for unaffected and affected labelled joints using FOIAS (k = 3).** (A) Comparison between affected and unaffected labelled joints (sub-category: affected, k = 3). (B) Comparison between score = 1 and unaffected labelled joints (sub-category: score = 1, k = 3). (C) Comparison between score = 2 and unaffected labelled joints (sub-category: score = 2, k = 3). (D) Comparison between score = 3 and unaffected labelled joints (sub-category: score = 3, k = 3).
(TIF)

**S5 Fig. Comparison between the calculated scores for unaffected and affected labelled joints using FOIAS (k = 5).** (A) Comparison between affected and unaffected labelled joints (sub-category: affected, k = 5). (B) Comparison between score = 1 and unaffected labelled joints (sub-category: score = 1, k = 5). (C) Comparison between score = 2 and unaffected labelled joints (sub-category: score = 2, k = 5). (D) Comparison between score = 3 and unaffected labelled joints (sub-category: score = 3, k = 5).
(TIF)

**S6 Fig. Comparison between the calculated scores for unaffected and affected labelled joints using FOIAS (k = 9).** (A) Comparison between affected and unaffected labelled joints (sub-category: affected, k = 9). (B) Comparison between score = 1 and unaffected labelled joints (sub-category: score = 1, k = 9). (C) Comparison between score = 2 and unaffected labelled joints (sub-category: score = 2, k = 9). (D) Comparison between score = 3 and unaffected labelled joints (sub-category: score = 3, k = 9).
(TIF)

## Acknowledgments

We would like to thank the participating patients and study centre personnel as well as Ulf Henkemeier, Lorenz Sparrenberg, Benjamin Greiner and Andrea Esser for valuable discussions.

## Author Contributions

**Conceptualization:** Lukas Zerweck.

**Data curation:** Michaela Köhm, Gerd Geißlinger, Frank Behrens.

**Formal analysis:** Lukas Zerweck, Phuong-Ha Nguyen.

**Funding acquisition:** Michaela Köhm, Andreas Pippow.

**Methodology:** Lukas Zerweck, Phuong-Ha Nguyen.

**Project administration:** Michaela Köhm, Phuong-Ha Nguyen, Andreas Pippow.

**Software:** Lukas Zerweck.

**Writing – original draft:** Lukas Zerweck, Michaela Köhm.

**Writing – review & editing:** Phuong-Ha Nguyen, Gerd Geißlinger, Frank Behrens.

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
