## [Decision Letter · Decision Letter 0]

15 Oct 2021

PONE-D-21-05449

An objective, automated and robust scoring using fluorescence optical imaging to evaluate changes in micro-vascularisation indicating early arthritis

PLOS ONE

Dear Dr. Zerweck,

Thank you for submitting your manuscript to PLOS ONE. After careful consideration, we feel that it has merit but does not fully meet PLOS ONE’s publication criteria as it currently stands. Therefore, we invite you to submit a revised version of the manuscript that addresses the points raised during the review process.

We look forward to receiving your revised manuscript.

Kind regards,

Wei-Yen Hsu, Ph.D.

Academic Editor

PLOS ONE

Journal Requirements:

2. In your Methods, please provide a description of how patients were recruited to your study.

4. Thank you for stating the following financial disclosure: "The authors declare that there was funding from the Fraunhofer Excellence Cluster for Immune mediated diseases  (https://www.cimd.fraunhofer.de/en.html) . The funders had no role in study design, data collection and analysis, decision to publish, or preparation of the manuscript."

We note that one or more of the authors is affiliated with the funding organization, indicating the funder may have had some role in the design, data collection, analysis or preparation of your manuscript for publication; in other words, the funder played an indirect role through the participation of the co-authors. If the funding organization did not play a role in the study design, data collection and analysis, decision to publish, or preparation of the manuscript and only provided financial support in the form of authors' salaries and/or research materials, please do the following:

a. Review your statements relating to the author contributions, and ensure you have specifically and accurately indicated the role(s) that these authors had in your study. These amendments should be made in the online form.

b. Confirm in your cover letter that you agree with the following statement, and we will change the online submission form on your behalf: 

“The funder provided support in the form of salaries for authors [insert relevant initials], but did not have any additional role in the study design, data collection and analysis, decision to publish, or preparation of the manuscript. The specific roles of these authors are articulated in the ‘author contributions’ section.

Reviewers' comments:

Reviewer's Responses to Questions

**Comments to the Author**

1. Is the manuscript technically sound, and do the data support the conclusions?

Reviewer #1: Yes

Reviewer #2: Yes

Reviewer #3: Yes

2. Has the statistical analysis been performed appropriately and rigorously? 

Reviewer #1: Yes

Reviewer #2: Yes

Reviewer #3: Yes

3. Have the authors made all data underlying the findings in their manuscript fully available?

Reviewer #1: Yes

Reviewer #2: Yes

Reviewer #3: Yes

4. Is the manuscript presented in an intelligible fashion and written in standard English?

Reviewer #1: Yes

Reviewer #2: Yes

Reviewer #3: Yes

5. Review Comments to the Author

Reviewer #1: The Authors describe a method of assessing FOIAS interpretation automatically, with hand crafted features and clustering algorithms combined. They show that as a distribution of mild and critical patients and of healthy individuals are distinguishable by the method.

However, in order to prove that the method is able to provide some probability of joints belonging into one or the other diagnostic set, it is advisable that the Authors perform a patient by patient, and/or joint by joint analysis in the spirit of the followings:

Discriminating between distributions based on their median or some other statistical summary technique with confidence intervals is not sufficient for diagnosis. However it is sufficient to show, that there is an effect in the cohort that indicates that the features and analytics describe diagnostically valid distinctions.

Instead I suggest they do a machine learning classificator over the features, using the labeling provided in Fig 7. e.g. for classes. What is important, that there is a cross-validation scheme employed as well, prediction results on test data should be the main results. Classificators will yield probabilities for each patient and/or joint and/or pixels belonging to each category of disease or healthy.

Other than that the study is very interesting and important. It is likely that the future works mentioned by the authors and others are necessary to achieve reasonable single patient/joint/pixel classification, and I highly encourage the authors to do these extra analysis now.

Reviewer #2: The manuscript proposes a method for automatically and objectively analyze the measured fluorescence intensity generated by Fluorescence Optical Imaging to evaluate the amount of inflammation of each of the subject’s joints focusing on the distinction between normal joint status or arthritis in psoriatic arthritis patients compared to healthy volunteers.

The research results show that the proposed methodology can present a tendency towards a classification into unaffected and inflamed joints, although wide research is required to transform that tendency into a robust classification model.

I find the topic interesting and being worth of investigation and the document is well strucutred, organized, fluidly written, the methodology followed is clearly explained, the results are clearly presented and support the conclusions.

Although I propose the following suggestions / considerations:

- I strongly suggest authors from refraining using personal pronouns such as "we" and "our" throughout the text and I encourage them to write it in an impersonal form of writing.

- How were the 12 healthy volunteers selected, are they statistically representative to act as controls for the sample of 163 patients used.

- Would the results improve if some artificial intelligence techniques such as machine learning classifiers or deep learning models are used, this could be addressed at the discussion.

- Only 6 out of 16 references are of the last 5 years, more recent relevant references are advised to be included.

Reviewer #3: The author have implemented the novel work entitled An objective, automated and robust scoring using fluorescence optical imaging to evaluate changes in micro-vascularisation indicating early arthritis" in a systematic way.But there are few queries need to be addressed.

1. Quantitative results need to be included in the result section of abstract.

2. The aim and objectives should be refined and need to be included at the end of the introduction section.

3. It was mentioned that "After 10 s the prepared colour agent Indocyanine green (ICG) is 75

injected into the patient with a dose of 0.1 mgICG/kgbody weight". which area of interest the dye is injected? Imaging is taken after how many hours of dye injection?

4. Whether dye is injected for healthy controls?

5. How do you quantitatively evaluate the micro vascularization in arthritis ? Do you have any parameters to describe?

6. Whether the images for arthritis has well as healthy subjects were displayed?

7.What does figure 2d represents? It seems to be figure caption given in the article and figure2 doesn't match

8. How do you match the concentric circles into the finger joints as given in figure 5? because the size of the finger joints will differ for each patient?

9. Discussion section is completely missing. Need to be included

6. PLOS authors have the option to publish the peer review history of their article (what does this mean?). If published, this will include your full peer review and any attached files.

Reviewer #1: No

Reviewer #2: **Yes: **RICARDO VARDASCA

Reviewer #3: No

---

## [Author Response · Author response to Decision Letter 0]

9 Feb 2022

Editor 1.)

Q: Please ensure that your manuscript meets PLOS ONE's style requirements, including those for file naming. The PLOS ONE style templates can be found at https://journals.plos.org/plosone/s/file?id=wjVg/PLOSOne_formatting_sample_main_body.pdf and https://journals.plos.org/plosone/s/file?id=ba62/PLOSOne_formatting_sample_title_authors_affiliations.pdf

A: The supporting information have been moved to the very end of the manuscript and are now following the references.

Editor 2.)

Q: In your Methods, please provide a description of how patients were recruited to your study.

A: Patients / volunteers were recruited from two prospective non-interventional studies (one with inclusion of patients diagnosed with psoriasis or psoriatic arthritis and one with inclusion of healthy volunteers without complaints in the joints of the hands).

This information was added to the subsection “Patient Group”.

Editor 3.)

Q: Please provide additional details regarding participant consent. In the ethics statement in the Methods and online submission information, please ensure that you have specified (1) whether consent was informed and (2) what type you obtained (for instance, written or verbal, and if verbal, how it was documented and witnessed). If your study included minors, state whether you obtained consent from parents or guardians. If the need for consent was waived by the ethics committee, please include this information.

A: All patients / volunteers provided signed informed consent for inclusion and were fully capable to give informed consent for participation in the study.

This information was added to the subsection “Patient Group”.

Editor 4.)

Q: Thank you for stating the following financial disclosure: "The authors declare that there was funding from the Fraunhofer Excellence Cluster for Immune mediated diseases (https://www.cimd.fraunhofer.de/en.html) . The funders had no role in study design, data collection and analysis, decision to publish, or preparation of the manuscript."

We note that one or more of the authors is affiliated with the funding organization, indicating the funder may have had some role in the design, data collection, analysis or preparation of your manuscript for publication; in other words, the funder played an indirect role through the participation of the co-authors. If the funding organization did not play a role in the study design, data collection and analysis, decision to publish, or preparation of the manuscript and only provided financial support in the form of authors' salaries and/or research materials, please do the following:

a. Review your statements relating to the author contributions, and ensure you have specifically and accurately indicated the role(s) that these authors had in your study. These amendments should be made in the online form.

b. Confirm in your cover letter that you agree with the following statement, and we will change the online submission form on your behalf: 

“The funder provided support in the form of salaries for authors [insert relevant initials], but did not have any additional role in the study design, data collection and analysis, decision to publish, or preparation of the manuscript. The specific roles of these authors are articulated in the ‘author contributions’ section.

A: Option B was chosen and the statement was added to the cover letter.

Editor 5.)

Q: We note that you have stated that you will provide repository information for your data at acceptance. Should your manuscript be accepted for publication, we will hold it until you provide the relevant accession numbers or DOIs necessary to access your data. If you wish to make changes to your Data Availability statement, please describe these changes in your cover letter and we will update your Data Availability statement to reflect the information you provide.

A: The data was uploaded and is accessible. The DOI is:

10.5281/zenodo.5705208

And the target url:

https://doi.org/10.5281/zenodo.5705208

Editor 6.)

Q: Please include your full ethics statement in the ‘Methods’ section of your manuscript file. In your statement, please include the full name of the IRB or ethics committee who approved or waived your study, as well as whether or not you obtained informed written or verbal consent. If consent was waived for your study, please include this information in your statement as well. 

A: Ethical approval was received from the ethics committee of the University Hospital Frankfurt a. Main, Germany. All patients / volunteers provided signed informed consent.

This information was added to the subsection “Patient Group”.

Reviewer #1 1.)

Q: I suggest they do a machine learning classificator over the features, using the labeling provided in Fig 7. e.g. for classes. What is important, that there is a cross-validation scheme employed as well, prediction results on test data should be the main results. Classificators will yield probabilities for each patient and/or joint and/or pixels belonging to each category of disease or healthy.

A: As suggested a machine learning classifier (random forest) for the clinical label as well as the FOIAS label has been trained and tested.

For both cases classes are not balanced. Therefore, all class sizes have been cropped to the smallest class size (for example for the FOIAS classifier the number of score 3 joints is 18). For each run samples from the classes containing more than the minimum amount of examples have been picked randomly. Due to the cropping many samples remain unconsidered for each run. Thus, the classifiers have been trained and tested 500 times.

Since for the clinical labeling a joint can be labeled as healthy, swollen, tender or swollen and tender the classifier has been trained to distinguish only between unaffected or affected scores. Thus, a binary classification problem has to be solved. The calculated F1 scores range from around 40 % to around 62 % detection rate.

For the FOIAS labeling a 4 class (Score 0, 1, 2 or 3) classification problem has to be solved. The calculated F1 scores range from 0 % to 66 % detection rate.

These fluctuating results are based on the overlap of scores for unaffected and affected joints and motivate to understand the causes of the heterogeneity of the data in more detail, since a classification with the current data seems extremely difficult. Therefore, we are currently conducting a study investigating the non-disease related impact factors on the signal (for example temperature, BMI, alcohol consumption and many more).

A subsection describing the machine learning approach has been added as well as a paragraph in the discussion describing the results.

Reviewer #2 1.)

Q: I strongly suggest authors from refraining using personal pronouns such as "we" and "our" throughout the text and I encourage them to write it in an impersonal form of writing.

A: The relevant passages have been changed.

Reviewer #2 2.)

Q: How were the 12 healthy volunteers selected, are they statistically representative to act as controls for the sample of 163 patients used.

A: The 12 healthy volunteers were selected with regards to the following in- and exclusion criteria: (a) missing musculoskeletal complaints on the hands, (b) missing diagnosis of joint diseases, (c) missing comorbidities with focus on diseases that go ahead with joint diseases, (d) missing contraindications to use indocyanine green as colour agent for FOI examinations. Due to these criteria, the 12 volunteers are not affected by PsA and can act as a control group.

This passage has been added to the manuscript.

Reviewer #2 3.)

Q: Would the results improve if some artificial intelligence techniques such as machine learning classifiers or deep learning models are used, this could be addressed at the discussion.

A: Refer to reviewer 1. 

Reviewer #2 4.)

Q: Only 6 out of 16 references are of the last 5 years, more recent relevant references are advised to be included.

A: We would have liked to include more recent works. But, to our knowledge there have not been many approaches to automatically evaluate the FOI images in the recent years. Additionally, in this work we have not been using machine learning approaches in the computer vision domain. In our current research, we shift our focus to a more machine learning based approach (especially for the segmentation task), which will also lead to more recent references.

Reviewer #3 1.)

Q: Quantitative results need to be included in the result section of abstract.

A: In this work a method to possibly overcome the heterogeneity in the data is presented and a tendency towards this goal was achieved. This result is not expressed in numbers. 

Reviewer #3 2.)

Q: The aim and objectives should be refined and need to be included at the end of the introduction section.

A: The last paragraph of the introduction has been rephrased to emphasis the aim and objectives of the study.

Reviewer #3 3.)

Q: It was mentioned that "After 10 s the prepared colour agent Indocyanine green (ICG) is 75 injected into the patient with a dose of 0.1 mgICG/kgbody weight". which area of interest the dye is injected? Imaging is taken after how many hours of dye injection?

A: The dye is injected into the arm (it was added in the manuscript).

The dye is injected 10 s after the Xiralite machine starts taking the images. Thus, the images are taken while adding the dye.

Reviewer #3 4.)

Q: Whether dye is injected for healthy controls?

A: Yes, the dye is the fluorescent substance and thus, essential for each measurement. For clarification, the word “patient” was changed to the word “proband” in the “Data acquisition” section and one more sentence mentioning this was added.

Reviewer #3 5.)

Q: How do you quantitatively evaluate the micro vascularization in arthritis ? Do you have any parameters to describe?

A: Clinical examination can be used to detect signs of changes in vascularization in arthritis that go along with specific signs of inflammation (including tumor, calor, dolor, rubor, function laesa). A reliable method to measure changes in micro-vascularisation that is not invasive such as angiography which is limited by x-ray load is not available in clinical routine care. So, we used clinical examination (joint assessment) as control assessment for the comparison between the groups.

Reviewer #3 6.)

Q: Whether the images for arthritis has well as healthy subjects were displayed?

A: Throughout, the presented work all images (Figure 1, 2, 4 and 5) refer to the same exemplary patient. This patient is not a healthy volunteer. Adding images of a healthy volunteer would not add any additional information.

Reviewer #3 7.)

Q: What does figure 2d represents? It seems to be figure caption given in the article and figure2 doesn't match

A: Figure 2d represents the extracted joint areas as a binary image. Figure 5 visualizes an overlay of the heatmap with these areas.

Reviewer #3 8.)

Q: How do you match the concentric circles into the finger joints as given in figure 5? because the size of the finger joints will differ for each patient?

A: Three different cases have to be differentiated:

1.) Joints located within in the fingers (DIP, PIP, IP):

The radius is defined as the smallest distance to the edge of the binary hand mask. Thus, the radius differs due to the width of the finger at the given location.

2.) Joints located within the hand (MCP), but omitting the Thumb’s MCP joint:

The radius is defined as the smallest distance to either the edge of the binary hand mask or the distance to the nearest other MCP location. Thus, these four MCP circles are matched to the hand’s width and to the individual hand

3.) Thumb’s MCP joint:

The size of this joint is equal to the size of the Thumb’s IP joint.

Reviewer #3 9.)

Q: Discussion section is completely missing. Need to be included

A: The results are discussed in the section “Evaluation and conclusion”. However, the chapter’s name was renamed for a clearer distinction.

---

## [Decision Letter · Decision Letter 1]

1 Sep 2022

An objective, automated and robust scoring using fluorescence optical imaging to evaluate changes in micro-vascularisation indicating early arthritis

PONE-D-21-05449R1

Dear Dr. Zerweck,

We’re pleased to inform you that your manuscript has been judged scientifically suitable for publication and will be formally accepted for publication once it meets all outstanding technical requirements.

Kind regards,

Diego Raimondo

Academic Editor

PLOS ONE

Additional Editor Comments (optional):

Reviewers' comments:

Reviewer's Responses to Questions

**Comments to the Author**

1. If the authors have adequately addressed your comments raised in a previous round of review and you feel that this manuscript is now acceptable for publication, you may indicate that here to bypass the “Comments to the Author” section, enter your conflict of interest statement in the “Confidential to Editor” section, and submit your "Accept" recommendation.

Reviewer #2: All comments have been addressed

Reviewer #3: All comments have been addressed

2. Is the manuscript technically sound, and do the data support the conclusions?

Reviewer #2: Yes

Reviewer #3: Yes

3. Has the statistical analysis been performed appropriately and rigorously? 

Reviewer #2: Yes

Reviewer #3: Yes

4. Have the authors made all data underlying the findings in their manuscript fully available?

Reviewer #2: Yes

Reviewer #3: Yes

5. Is the manuscript presented in an intelligible fashion and written in standard English?

Reviewer #2: Yes

Reviewer #3: Yes

6. Review Comments to the Author

Reviewer #2: The manuscript aims to automatically and objectively analyze the measured fluorescence intensity generated by Fluorescence optical imaging technique to evaluate the amount of inflammation of each of the subject’s joints focusing on the distinction between normal joint status or arthritis in psoriatic arthritis patients compared to healthy volunteers.

With the proposed method a tendency towards a classification into unaffected and inflamed joints can be achieved, but further research is necessary to transform the tendency into a robust classification model.

I find the topic interesting and being worth of investigation and the document is well strucutred, organized, fluidly written, the background is adequate, the methodology well explained (formulas are correct), results are clearly presented, supporting the discussion and conclusions.

I am happy with the authors' answers and action towards the reviewers questions and comments and significantly have improved the manuscipt which I support its acceptance for publication at Thermal Biology journal.

Reviewer #3: The authors addressed the queries raised by the reviewer. Hence the article can be accepted in its current form.

7. PLOS authors have the option to publish the peer review history of their article (what does this mean?). If published, this will include your full peer review and any attached files.

Reviewer #2: No

Reviewer #3: No

---

## [Editor Report · Acceptance letter]

6 Sep 2022

PONE-D-21-05449R1 

An objective, automated and robust scoring using fluorescence optical imaging to evaluate changes in micro-vascularisation indicating early arthritis 

Dear Dr. Zerweck:

I'm pleased to inform you that your manuscript has been deemed suitable for publication in PLOS ONE. Congratulations! Your manuscript is now with our production department. 

Kind regards, 

on behalf of

Dr. Diego Raimondo 

Academic Editor

PLOS ONE